# Exotic Grasses Reduce Infiltration and Moisture Availability in a Temperate Oak Savanna

**DOI:** 10.3390/plants11192577

**Published:** 2022-09-30

**Authors:** Ze’ev Gedalof, Lesley E. Davy, Aaron Berg

**Affiliations:** 1Climate & Ecosystem Dynamics Research Laboratory, Department of Geography, Environment and Geomatics, University of Guelph, Guelph, ON N1G 2W1, Canada; 2Department of Geography, Environment and Geomatics, University of Guelph, Guelph, ON N1G 2W1, Canada

**Keywords:** *Quercus garryana*, soil moisture, temperate oak savannas, seedling recruitment, competition, invasive species

## Abstract

Biological invasions represent one of the most urgent conservation challenges. Oregon white oak (*Quercus garryana*) savannas, a complex of grassland and transitional forest, are especially sensitive to these invasions. These ecosystems have been severely degraded and fragmented over the past century and are being encroached by conifers, and oak seedlings are failing to emerge from the understory at many locations. Understanding competitive interactions between Oregon white oak and associated native and exotic vegetation would provide insight into forest-grassland dynamics and the role of exotic grasses in the decline of native species, the processes that maintain temperate savanna ecosystems, and the role of soil water uptake by individual savanna species in contributing to overall species assemblages. In this study, we quantified the soil moisture budget for invaded and uninvaded oak-associated ecosystems. From February to October 2007 we used a split paired plot experiment in Duncan, British Columbia, Canada to measure soil moisture on treatment sites where exotic grasses were removed with herbicide and control plots where they were not, using three depths (5, 20, and 35 or 50 cm) in the soil profile. Our results show that the plots that contained exotic vegetation had a faster rate of soil drying following precipitation events at the 5 cm depth than plots with the predominantly native species. We attribute this difference to the capacity of exotic vegetation to exploit soil moisture more rapidly than native vegetation at times of the year when native vegetation cannot. These results provide insight into one mechanism by which exotic grasses affect associated native plants and could help guide restoration efforts.

## 1. Introduction

Biological invasions are among the most urgent conservation challenges [1]. However, the processes by which exotic species come to dominate their recipient ecosystems are poorly understood, in part because the plant communities predicted by most candidate hypotheses are similar to each other in terms of their predicted structure and composition in response to invasions [2]. There are several competing hypotheses to explain the success of invaders, e.g., as the authors of [3] propose, but they can be broadly placed into one of two categories based on the relative importance of interactions with their associates: In strong interactive models, communities are structured by the direct effects of dominants on their associates through processes, such as competition, allelopathy, and herbivory [2,4]. In weak interactive models, dominance occurs because exotic species, for one reason or another, are less recruitment-limited than their native associates. In this case, exotics dominate because they are better-suited to environmental conditions, such as soil conditions or liberation from predators (e.g., as in [5]) compared to their native associates.

Temperate oak savannas are among the most highly degraded ecosystems in western North America due to a suite of interacting processes, including fire suppression, grazing, logging, and species introductions [6,7,8], though eastern oak savannas are also often impacted [9,10]. Observed recent changes in these systems include a widespread lack of recent recruitment of oak to the overstory [10,11,12,13], encroachment by shade-tolerant tree species [7,12,13,14], and increases in exotic species; in particular, these include agronomic grasses, such as orchard grass (*Dactylis glomerata*), sweet vernal grass (*Anthoxanthum odoratum*), barren brome (*Bromus sterilis*), and colonial bentgrass (*Agrostis capillaris*) [6,13,15]. The strong response of these ecosystems to perturbations suggests that they provide an suitable model system for understanding the mechanisms of species coexistence and the effects of environmental change on ecosystem structure and composition. Most previous studies have focused on methods of invasive species control or eradication without an understanding of the mechanisms by which those species become dominant (e.g., see [3,6,16]).

Oregon white oak (*Quercus garryana* Dougl. ex Hook.) savannas, a complex of grassland and transitional forest, have especially been impacted by species introductions and fire exclusion [6,17,18]. Oregon white oak associated ecosystems span an extensive north–south range, from Vancouver Island and Southern British Columbia (where the species is known as Garry oak), through the valleys and foothills of Washington and Oregon, and into montane regions in Southern California [19,20,21]. These ecosystems have been severely degraded and fragmented over the past century and many are characterized by a lack of overstory recruitment by Oregon white oak over majority of the last 50 years [13,22,23]. The introduction of invasive species has also posed a serious problem to Oregon white oak associated ecosystems: over 80 percent of herbaceous cover may be composed of non-native species [6,24]. Exotic grasses, such as orchard grass (*Dactylis glomerata*) and sweet vernal grass (*Anthoxanthum odoratum*) may comprise over 30 percent of the total vegetation cover in Garry-oak-associated ecosystems [6,13,25]. The rapid spread of Scotch broom (*Cytisus scoparius*) has also replaced native plants, changed soil nutrients, and dramatically altered the composition of these ecosystems [6,25]. Invasive insects such as spongy moth (*Lymantria dispar*) and winter moth (*Operophtera brumata*) are occasional predators of mature trees in British Columbia but are rarely lethal (Stein 1990). Recently, powdery mildew, a complex of invasive microorganisms, has been proposed as a potential cause of recruitment failure in oak [26], though, in British Columbia they have not been reported to be lethal to Oregon white oak.

Understanding the mechanisms by which exotic species have come to dominate these ecosystems and the effects that they have on native species should provide insight into the processes that structure oak savannas and provide managers with tools to use for the restoration of invaded ecosystems. While a lack of successful recruitment by oak is likely a consequence of a suite of interacting variables [27], in savanna environments, water is often implicated as the resource that is most effectively depleted by herbaceous species [28,29,30]. Thus, displacement of native plants and the poor survivorship of oak seedlings may be a consequence of changing water relations in temperate oak savannas [28,31,32]. Gordon and Rice [28] have suggested that annual grassland species are more likely to rapidly deplete soil water in the upper soil layers during the spring compared to the associated native perennial species (e.g., see [33,34]). Earlier phenologies of introduced species and greater density and cover in annual species over perennial grassland species may result in the rapid depletion of soil water [28,35]. Gordon and Rice [28] also found that seedling shoot emergence was most inhibited when high densities of annual species induced rapid rates of soil water depletion. Consequently, soil moisture deficits should have the greatest impact on oak seedlings and their associates during the spring and summer dry months, characteristic of the Mediterranean climate regime that is typical throughout the range of Oregon white oak [32].

The goal of the current research was to test the hypothesis that introduced annual grasses exploit soil moisture differently from oak seedlings and native associates, potentially explaining their reduced establishment and cover. Specifically, we predicted that introduced grasses exploit soil moisture earlier in the year than associated natives and do so further up in the soil profile. To identify this effect, we looked at both mean soil moisture values and the effect of exotic grasses on soil moisture following summer rainfall events, when water is most limiting to growth.

## 2. Materials and Methods

This hypothesis was tested using a split paired plot experiment with predominantly native versus mixed native and exotic vegetation assemblages at the Cowichan Garry Oak Preserve in Duncan, British Columbia. This preserve is one of the largest and most diverse Oregon white oak savanna ecosystems remaining in Canada (48°48′ N, 123°37′ W). The preserve is a 10.80 ha plot and consists of approximately 60 percent mixed forest and 40 percent open grassland (Land Trust Alliance of BC 2002). Climate within the range of Oregon white oak is broadly Mediterranean, with warm dry summers and cool wet winters. Climate normals for Duncan, BC (48°44′ N, 123°4′ W, 1981–2010) are consistent with this: mean Summer temperatures (April–September) are 14.6 °C and winter temperatures (October–March) are 5.6 °C. Summer and winter precipitation totals are 42.6 mm and 184.3 mm, respectively.

## 3. Plot Preparation and Selection

Soil moisture was monitored at three depths (described below) in two sets of paired plots, for a total of three probes each at four plots, that are part of the “Regional Strategies for Restoring Invaded Prairies” project (Stanley et al., 2008; 2011A, 2011B). We selected two 5 × 5 m plots where exotic grasses had been eradicated (treatment) and two 5 × 5 m adjacent plots where they had not (control). Exotic grasses were removed by applying a grass-specific herbicide (sethoxydim; see Stanley et al., 2011B for additional details) in the spring of 2005 and 2006, and then subsequently mowed once a year to reduce the limiting effects of thatching [2] (Stanley et al., 2008). The herbicide used kills grass species during their early stages of development. Most critically for this study, native grasses at the preserve have been found to be resistant to the application of the herbicide at the time of year it was applied, presumably due to early-season phenological differences between native and exotic grasses (Tim Ennis, Director of Land Stewardship, Nature Conservancy of Canada, personal communication, 18 January 2006). This combined treatment reduces the vigor of the exotic grasses relative to the native vegetation, inhibits grass seed development, and minimizes the effects of thatching on subsequent seed germination and water infiltration [36,37,38].

Control plots were established adjacent to each of the treatment plots; these plots had both exotic and native species present. The plots are numbered according to the system used by Stanley et al. [37] in order to be consistent with other publications that have emerged from the “Regional Strategies for Restoring Invaded Prairies” project. Consequently, the two control plots are numbered 2 and 14, and their corresponding treatment plots are numbered 3 and 15. Plots 2 and 14 have approximately 30 and 45 percent exotic grass cover, respectively (primarily *Dactylis glomerata*, *Poa pratensis*, and *Bromus sterilis*); plots 3 and 15 have less than 1 percent exotic grass cover. The dominant native vegetation on these plots was primarily *Dodecatheon hendersonii*, *Lomatium utriculatum*, and *Sanicula crassicaulis*. All plots had a small number of Oregon white oak seedlings (total cover <1%), and were proximal to mature Oregon white oak trees with a canopy closure of approximately 25%.

## 4. Monitoring Soil Moisture Variability

Soil moisture was measured using Steven’s Hydra Probes^TM^ that were installed at each of the four plots. One hole was dug in each of the 5 m by 5 m plots and the probes were installed at 5 cm, 20 cm, and 50 cm depths (except in Plot 2, where the bedrock was nearer to the surface and the probe was installed at a 35 cm depth; in all cases, the bottom probes were below the effective rooting depth of the understory vegetation). In order to minimize the impact of instrument installation on the soil profile, the metal tines of the probe were directly inserted into the undisturbed face of the exposed soil. The hole itself was also dug with the smallest diameter that still allowed for installation of the instruments. Soil horizons were kept separate and returned to their original sequence within the profile. Vegetation and surface litter overlying the hole was removed as an intact mass and was returned following installation of the probes with no subsequent mortality observed. While the holes dug for installation of the equipment could change the hydraulic properties of the study site, the same installation process was used for all four probes, so results should be directly comparable.

The Hydra Probe instrument measures the dielectric properties of the soil, which is then used to determine the soil water content. For this study, the deployed Hydra Probes were used to determine soil water and temperature every four hours from February to October 2007. The six installed Hydra Probes were attached to a data logger, a battery, and a solar panel for each split paired plot.

Each Hydra Probe was individually calibrated so that the real dielectric constant originally measured by the probes could be standardized to the volumetric water content to allow direct comparisons between plots. To complete the calibration of the probes we used soil samples taken from each of the plots at Cowichan. A representative sample of the sampling region of each probe was placed into a plastic container and a template resembling Hydra Probe was inserted into each sample following the procedure detailed in Burns et al. [39]. As provided from the manufacturer and using soil specific equations, the accuracy of the Hydra Probe for computing soil water content is estimated at ± 0.03 (water fraction by volume); however, the development of soil-specific calibration coefficients reduces the measurement uncertainty to below +/−0.02 (water fraction by volume), e.g., as in [39,40,41].

This template ensures that the metal tines of the Hydra Probe can be inserted into the oven -ried sample. The initial wet sample was then dried in an oven for 24 h at 80 °C. The Hydra Probe was then inserted into the oven-dried soil and the soil was slowly re-wetted over a 12 h period using a very slow but continuous drip rate from an intravenous bag. Continuous measurement of the soil mass (as measured from a recording weight scale) and dielectric response were recorded for calculation of the soil-specific calibration curve that relates the measured soil dialectic response to soil water content (m^3^ m^−3^).

## 5. Quantifying Soil Moisture Differences

Meteorological data for temperature and precipitation were obtained from Duncan Kelvin Creek meteorological station, British Columbia (48°44′ N, 123°43′ W) for the period from February to November 2007. The meteorological station is approximately 11 km from the preserve, and is located within the same valley and at a comparable altitude, so it should be highly representative of the site. Daily data were used for all analyses.

To identify rainfall events, it was necessary to first determine the volume of precipitation necessary to have an impact on soil moisture as observed at our sensors installed at 5 cm. We denote this quantity as our effective input. Based on a comparison of precipitation events and observed soil moisture responses, we determined that precipitation events in excess of 9 mm caused saturation of the soil at the location of the 5 cm sensor. Saturation near the surface was a necessary initial condition in this case to allow pairwise comparisons of the control and treatment plots, as conditions less than saturation would not allow for site-to-site comparisons. If more than one rain event was recorded within a four-day period, the terminal rain event was used to determine the rate of subsequent drying of soil among the plots. Prior to 1 May and after 31 October, soil moisture at both pairs of plots was found to be continuously near saturation; therefore, water is not expected to be a limiting factor at this time of the year. For this reason, the dry down rates were calculated only for the period from May to October.

Following the identification of our effective input events, the drying rate was computed by calculating the percent difference of the soil water content at saturation to the soil water content observed after 1, 2, and 4 days following the saturation event. We chose to examine percent change in soil moisture rather than absolute volumetric water content so that differences would not be affected by the very strong seasonal cycle in soil moisture imposed on the record by the Mediterranean climate (see Figure 1). For each effective input event an average drying rate was calculated for each of the two control and two treatment plots. Differences in the mean drying rates of control and treatment plots for all events over the growing season were tested for significance using a paired samples *t*-test (8 events times 2 replicates, for a total sample size of 16 in each of the control and treatment calculations). Soil textures, slope, and canopy closure were nearly identical between the control and treatment pairs, allowing for direct comparison of soil water properties between paired plots.

## 6. Results

### 6.1. Precipitation

Daily precipitation was analyzed for the period from 1 May to 2 October 2007. Eight discrete rain events were identified over the study period, terminating on 20 May, 9 June, 28 June, 21 July, 3 September, 18 September, 30 September, and 19 October (Figure 1).

### 6.2. Seasonal Soil Moisture Curves

Subtracting the soil moisture curve of each control plot from its associated treatment shows the effect of the presence of exotic grasses on soil moisture availability (Figure 2 and Figure 3). There was positive agreement between the two pairs of plots in terms of this effect: At the 5 cm depth, soil moisture was 6.1% and 3.5% higher in the two plots with exotic grasses present compared to their respective treatment plots. At the 20 cm depth, the opposite pattern was seen, with soil moisture at 11.0 and 4.6 percent lower in the presence of exotic grasses. At the bottom of the profiles (below the effective rooting depth seen in any of the plots), there was no significant difference between the control and treatments (2.0 and 1.1 percent differences). Integrated over the depth of the soil profile, the net effect of the presence of the exotic grasses is lower than the volumetric water content during the growing season (paired *t*-test, *p* < 0.001 for both pairs of plots).

### 6.3. Soil Drying Rates after Precipitation Events

The dry down rate was calculated as the percent change in volumetric water content at each of the control and treatment plots for one, two, and four days following each of the eight precipitation events identified among both control and treatment sites. Site-wide mean drying rates were determined for both control plots and both treatment plots at 5 cm and 20 cm depths. At the 5 cm depth, the control plots had a significantly faster drying rate following rainfall events compared to the treatment (Figure 4). For one day after the precipitation event, the control drying rate was 1.18 percent compared to the treatment’s 0.52 percent (paired samples *t*-test, *p* = 0.003). The two-day drying rate for the control and treatment were 1.95 percent and 0.80 percent, respectively (*p* < 0.001). Lastly, the four-day drying rate at the 5 cm depth was 3.36 percent for the control and 1.53 percent for the treatment (*p* < 0.001). No significant differences between the control and treatment plot were found at the 20 cm depth (*p* > 0.05 in all cases; Figure 5).

## 7. Discussion

Three main effects of exotic grasses on volumetric soil moisture were seen in this experiment: First, plots with exotic grasses had lower overall soil moisture volumes during the growing season. Second, this water was unequally distributed within the soil profile. Plots with exotic grasses had more water present near the soil surface but less water present at lower depths. Third, the near-surface soil water was depleted more rapidly in plots with exotic grasses present than in those without. Together, these results suggest that rainfall that occurs upon plots with exotic grasses does not deeply infiltrate into the soil profile and is either used by vegetation or is directly evaporated from the thatch layer or the soil surface.

There are several processes that could be driving this pattern, as supported by the data collected. First, thatching associated with exotic grasses could be enhancing evaporation from intercepted water directly from the thatch layer. In this case, the thatching would increase interception and therefore decrease the total water available for infiltration into the deeper soil profiles. We think this process is unlikely to be important in this analysis, though, as the thatch layers observed were less than the 13 mm figure widely cited by turf managers as the threshold thickness needed to impede infiltration (e.g., Harivandi 1984). Furthermore, thatch has been found to affect only initial infiltration rates; sustained infiltration rates are unaffected by the presence of thatch [36] and the presence of grass roots may actually increase piping action, accelerating infiltration where they are present in the soil profile [42]. Additionally, if this process is important, the soil moisture levels seen in the control plots should have been lower at all levels compared to those seen in the treatment plots, which they were not.

Second, once infiltration into the upper soil profile has occurred, the greater shade offered by the exotic grasses relative to the native vegetation may help to maintain higher soil moisture levels by reducing insolation at the soil surface and thereby reducing evaporative losses. Given the lower overall moisture levels seen in the control plots, though, and that the soil moisture differences were opposite in sign between the near-surface and mid-profile depths, this process alone cannot explain the differences seen here.

A third process that might explain this pattern is that water that infiltrates the soil is rapidly transpired by the exotic grasses that either remain metabolically active after the native vegetation has senesced or green-up in response to the available moisture. Together, these processes point to a model in which rain penetrates the upper portion of the soil profile due to the piping action of grass roots, resulting in the higher soil moisture levels seen at the 5 cm depth, but is then transpired by grasses, precluding it from infiltrating deeper into the soil profile, resulting in the lower soil moisture levels seen at the 20 cm depth.

The soil moisture values seen at the 50 cm were similar between the pairs of plots, which may be attributable to lateral groundwater flow. Soil depths range from approximately 20–85 cm across the site and soils are coarse-textured, indicating that hydrologic conductivity is relatively high. Virtually no roots from understory plants penetrated this deep in the soil profile, so this water probably has little effect on understory dynamics.

In the context of our original research question, we conclude that the presence of exotic grasses may facilitate the infiltration of precipitation into the upper portions of the soil profile but this water does not subsequently contribute to summertime ground water recharge and is readily transpired by these grasses. In the dry summer climate of southern coastal British Columbia, this water probably contributes, in an important fashion, to the survival of Oregon white oak seedlings and their associates. These seedlings have shallow, poorly developed root systems and, unlike most of their understory associates, are photosynthetically active during summer months. The summer during which this study was undertaken was wetter than average: From 1 May to 31 October, there was 310 mm of total precipitation versus an average of 292, and there were 69 days with precipitation >= 0.2 mm versus an average of 59. Given that this study was undertaken during a particularly wet summer and dry down effects were still detected, we expect that these effects would be more pronounced during dry years (however, see [43]).

Previous research undertaken at this site found that weak interactive effects were probably most important in structuring communities in systems associated with Oregon white oak [2]. The results here suggest that strong interactive effects may also be important in structuring communities: exotic grasses intercept and deplete soil moisture more efficiently and to lower overall levels than their native associates, and they do so at a time of year when at least some of their associates are especially prone to soil moisture deficits. Whether these effects are the dominant process in structuring the resulting communities cannot be determined from this analysis and it is possible that weak interactive effects are more important in regulating the coexistence of grasses and forbs, which are metabolically active earlier in the season when water is less limiting. However, oak seedlings and many of their associates are metabolically active and require moisture in the upper soil profile during summer months and this moisture is reduced in the presence of exotic grasses. In spite of their fundamentally different functional types, competition for moisture between trees and grasses is probable. Gordon and Rice [28] reached similar conclusions for blue oak (*Quercus douglasii*): seedling establishment and growth rates were regulated by the presence of exotic grasses and their effect on soil moisture depletion rates. In their study, seedling mortality occurred whether soil moisture was reduced by exotic competitors or through direct manipulation, suggesting that moisture use by the exotic species was the cause of seedling mortality rather than other potential mechanisms [28].

As a foundation species in its ecosystem, Oregon white oak trees alter the surface microclimate and soil chemistry and provide a habitat structure to dozens of species. Any change to their dynamics almost certainly exerts indirect effects on their associated understory species. In the Pacific Northwest, oak savannas are especially susceptible to invasions due to their typical proximity to human settlements and open canopy structure. Furthermore, most climate models project increasing drought severity in western North America in the future, suggesting that competition for moisture will become increasingly important [44,45].

## 8. Conclusions

The split-paired plots at the Cowichan Garry Oak Preserve allowed us to measure the effects of exotic grasses on the soil moisture budgets of a temperate oak savanna ecosystem where exotic grasses were both present and experimentally removed. Exotic grasses were found to use water more efficiently than native species, leading to decreased soil moisture storage at lower depths and more rapid drying following rain events.

These results suggest that soil moisture dynamics play an important role in structuring temperate savannas and that disruptions to the soil moisture budget may contribute to savanna degradation. While weak interactive processes may be important in regulating community structure and composition, strong interactive processes have probably also contributed to the dominance of exotic grasses in these temperate oak savannas. Of particular importance is that strong interactive effects probably affect the survival of Oregon white oak seedlings and influence their recruitment to the overstory.

To minimize the effects of invasive grasses, we recommend active management where possible. The timely use of prescribed fire has been suggested to reduce grass cover, thatch, and seed production. Mowing is often a more acceptable substitute but it may contribute to thatch accumulation. Grazing has also been suggested as a restoration technique to both reduce cover and thatch, e.g., as in [38]. This research was originally motivated by the observation that Garry oak is failing to reproduce at most locations in British Columbia. The finding that exotic species exploit soil moisture at the same time of the year that oak is metabolically active suggests that their presence may play an important role in limiting recruitment. Similarly, the failure of oak to recruit at many locations may be indicative of more severe ecosystem disruptions at those locations.

## Figures and Tables

**Figure 1 plants-11-02577-f001:**
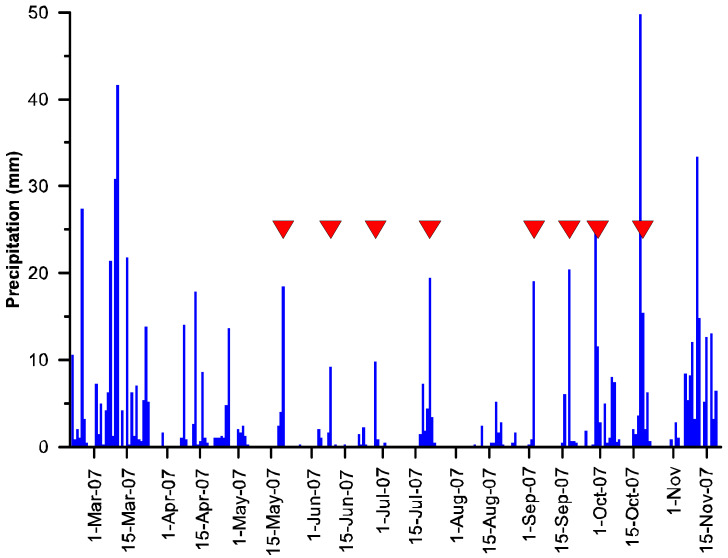
Precipitation observed at Kelvin Creek, Duncan, BC, Canada, from 20 February to 20 November 2007. The red triangles indicate effective input precipitation events considered in this analysis.

**Figure 2 plants-11-02577-f002:**
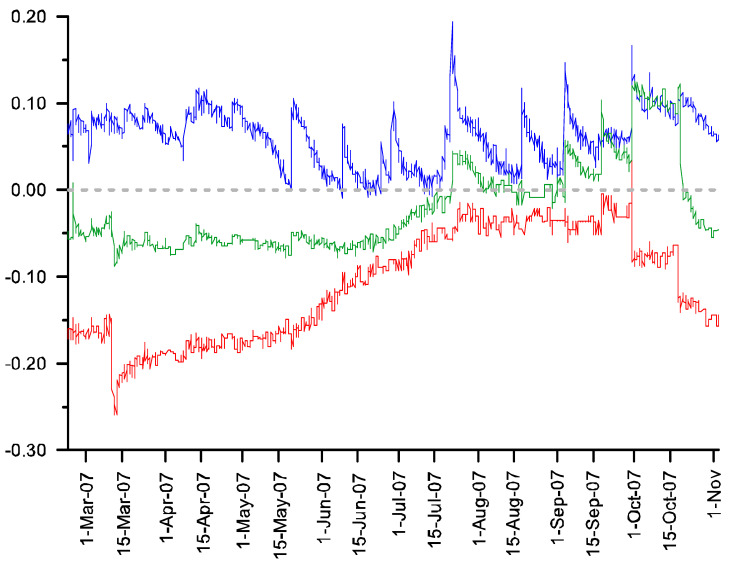
Differences in soil moisture between Plots 2 (control) and 3 (treatment) over time. The soil moisture probe depths are 5 cm (blue), 20 cm (red), and 35/50 cm (green). Positive values indicate that soil moisture is higher in presence of exotic vegetation.

**Figure 3 plants-11-02577-f003:**
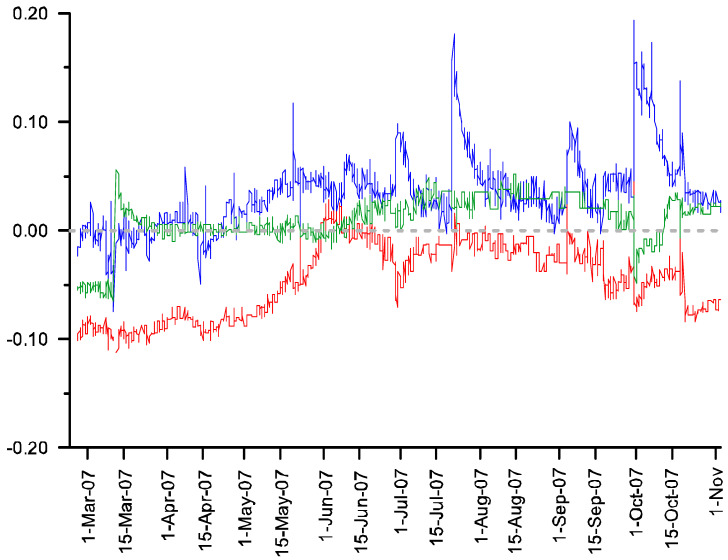
Differences in soil moisture between Plots 14 (control) and 15 (treatment) over time. The soil moisture probe depths are 5 cm (blue), 20 cm (red), and 50 cm (green). Positive values indicate that soil moisture is higher in presence of exotic vegetation.

**Figure 4 plants-11-02577-f004:**
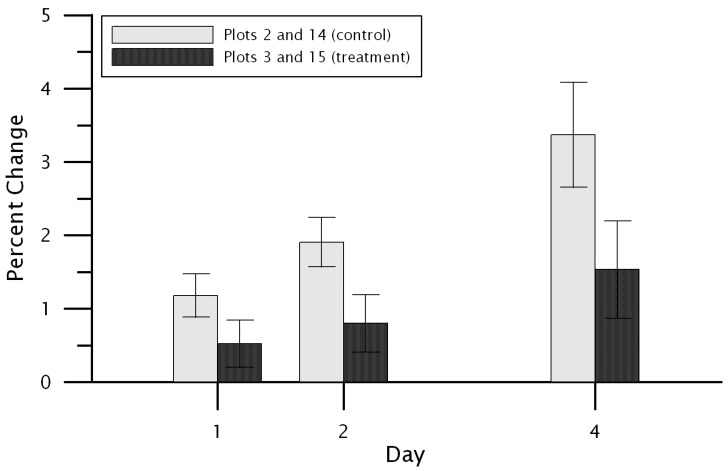
The mean drying rate for the 5 cm depth at the control plots (Plot 2 and Plot 14) and the treatment plots (Plot 3 and Plot 15). The drying rate is shown for 1, 2, and 4 days following precipitation events. Error bars indicate 95 percent confidence limits on the mean.

**Figure 5 plants-11-02577-f005:**
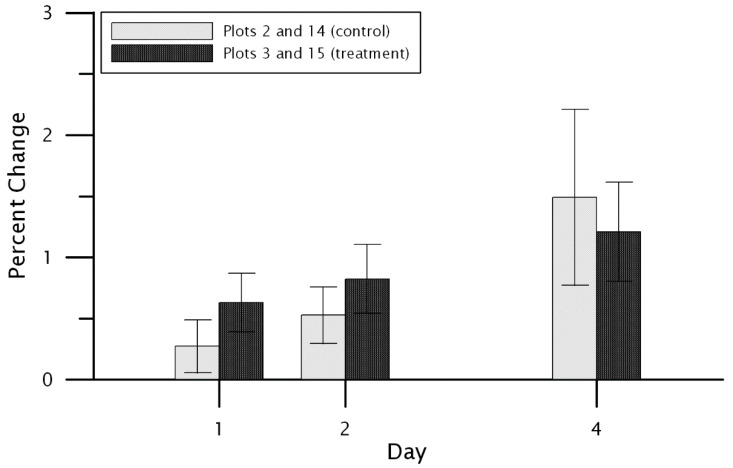
The mean drying rate for the 20 cm depth at the control plots (Plot 2 and Plot 14) and the treatment plots (Plot 3 and Plot 15). The drying rate is shown for 1, 2, and 4 days following precipitation events. Error bars indicate 95 percent confidence limits on the mean.

## Data Availability

Not applicable.

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
