# Peer review of "Exotic Grasses Reduce Infiltration and Moisture Availability in a Temperate Oak Savanna"

_plants, 2022, doi:10.3390/plants11192577_

Round 1

Reviewer 1 Report

The manuscript entitled ‘Exotic Grasses Reduce Infiltration and Moisture Availability in a Temperate Oak Savanna’ addresses and important and timely topic, exotic species contributing to major biodiversity losses, particularly in habitats already affected by other negative processes, such as the lack of overstory recruitment in the particularly sensitive oak savannas.

The general idea of the study and the methodology applied seem to be overall appropriate, but the presentation, in terms of scientific accuracy and quality of editing, does not have the fastidiousness one would expect from a manuscript targeting publication in Plants.

The abstract is not sufficiently informative. It is not clear what the treatments were, what the conditions in the control plots and which the three soil depths were.

The keywords partially repeat the title, while the first sentence partially repeats the abstract.

Line 4 is not clear enough: “the plant communities predicted by most candidate hypotheses are”, which are these hypotheses?

L 11: should be "through processes such as".

L 15: as these species are new to these environments, I would say that the local conditions are suitable for exotic species, rather that they are adapted to these conditions.

L 27: few scientific names for these agronomic grasses should be included.

L 32-33: this statement should be supported with further, incl. more recent references.

L 38-42: a map with the range and the study area within would be more informative.

L45-47: as mildew – an exotic invasive species (complex) – has been reported as a problem also in the case of the Oregon white oak, the authors should at least mention that it could be a contributing factor, if they’ve noticed such infestations in their study plots. The following review discusses this issue for Europe, describing several processes, which are similar:

Demeter et al. 2021. Rethinking the natural regeneration failure of pedunculate oak: The pathogen mildew hypothesis

L65-66: the importance of water supply could be discussed more in depth. Furthermore, the authors scarcely cite European studies, though there are several scientific findings relevant for the situation described for the Oregon white oak. The references below could help the authors elaborate more on this topic:

Annighöfer et al. Regeneration patterns of European oak species (Quercus petraea (Matt.) Liebl., Quercus robur L.) in dependence of environment and neighborhood.

Bobiec et al. 2018. Seeing the oakscape beyond the forest: a landscape approach to the oak regeneration in Europe.

Dreyer et al. 1991. Photosynthesis and shoot water status of seedlings from different oak species submitted to waterlogging.

Mendoza et al. 2009. A seeding experiment for testing tree-community recruitment under variable environments: implications for forest regeneration and conservation in Mediterranean habitats.

L102: the authors should inform the reader that the three depths are mentioned in the next section.

L123: Stanley et al. – is it 2008?

L125-130: a graph showing the arrangement of the plots would be more informative. It could be nicely merged on a figure with the map suggested earlier.

L131: what does “small number of Oregon white oak seedlings” mean?

Figures: The authors show their findings selectively, which is a very weak point of the manuscript. It is not clear why not all plot pairs are shown as subsets of the same graph. It would be more relevant than their difference.

eradicated – not eradicated – control: all three categories should be shown

L263-264, 270: it is not clear why only the control and the treatment plots are compared. Overall, the presentation of the findings is a bit difficult to follow. Informative graphs would be highly appreciated. Similarly, it is not clear which pairs of plots the author refer to in L326.

L333: the infiltration of soil moisture into – do the authors mean precipitation here?

The Discussion section should address more the findings of other studies, comparing these with the presented results. There are quite long sections without any references cited.

The Conclusions section should be concise, without any citations, but providing some management recommendations. Grazing for example could be used against these exotic species. L373-377 should be deleted. L390-391 are suitable for the discussion section, with references for similar studies, ex. those conducted in the Mediterranean dehesas.

Further suggested reference: Molinari et al. 2020. Where have all the wildflowers gone? The role of exotic grass thatch

Reviewer 2 Report

The authors present an interesting study about the related mechanisms in which invasive species dominate native ecosystems. In that case they study the Oregon White Oak savannas. Measuring the soil moisture budget at three depths (5, 20 and 35/50 cm) in invaded and non-invaded ecosystems the authors confirm that exotic grass species exploit soil moisture faster than native vegetation allowing the bio invasive processes. Despite the experiments were done in 2007, the results are still valid.

The study could interest to the audience and also provide, specially to environmental policy makers, methodologies and tools to restore invaded ecosystems.

Anyway, in order to become a paper for Plants journal I’ll suggest some changes.

General comments:

In the abstract you state that your research is going to involve two aspects; soil moisture budget for invaded and non-invaded ecosystems and seasonal phenology of white oak seedlings. Last research is not in the manuscript so you should take it out from here.   

The Latin name of the species should be placed in parenthesis. In some cases are in brackets.

The text cites should be numbered (journal guidelines).

Often, when you finish a sentence there are two points instead of one. Please check it carefully.

Detailed comments:

In page 2 (line 9), a point is necessary after “associate”. In line 35, put a parenthesi instead of bracket after “white oak”.

Page 3, in line 82 delete a couple of spaces. Same for lines, 94, 96 and 97.

Page 4, line 98 put the “W” of winter in lower case. Line 123, the cite of Stanley et al needs the year (anyway remember that cites should be numbered).

Figure 1. This figure is clear but maybe will be better to replace the red triangles of effective input precipitation events by another kind of symbol (maybe a small asterisk?).

Figure 2. In the caption of the figure erase a point at the end of the sentence.

Page 13. In line 390 point is needed between sentences.

References. Adjust to guidelines of the journal.

As you can see, globally there are minor changes and easy to resolve. Congratulations for the research I’ve enjoyed reading it.

Round 2

Reviewer 1 Report

The authors clarified the text of the manuscript, which is now improved.